# Double dome structure of the Bose–Einstein condensation in diluted $S = 3/2$ quantum magnets

Yoshito Watanabe [1] ✉, Atsushi Miyake [2], Masaki Gen[1], Yuta Mizukami[1], Kenichiro Hashimoto [1], Takasada Shibauchi [1], Akihiko Ikeda [3], Masashi Tokunaga [2,4], Takashi Kurumaji[1], Yusuke Tokunaga[1] & Taka-hisa Arima [1,4] ✉

Bose–Einstein condensation (BEC) in quantum magnets, where bosonic spin excitations condense into ordered ground states, is a realization of BEC in a thermodynamic limit. Although previous magnetic BEC studies have focused on magnets with small spins of $S \leq 1$, larger spin systems potentially possess richer physics because of the multiple excitations on a single site level. Here, we show the evolution of the magnetic phase diagram of $S = 3/2$ quantum magnet $Ba_2CoGe_2O_7$ when the averaged interaction $J$ is controlled by a dilution of magnetic sites. By partial substitution of Co with non-magnetic Zn, the magnetic order dome transforms into a double dome structure, which can be explained by three kinds of magnetic BECs with distinct excitations. Furthermore, we show the importance of the randomness effects induced by the quenched disorder: we discuss the relevance of geometrical percolation and Bose/Mott glass physics near the BEC quantum critical point.

Bose–Einstein condensation (BEC), in which a macroscopic number of bosons occupy the same state, has been studied broadly in areas ranging from condensed matter physics and particle physics to astrophysics. In condensed matter physics, a variety of systems such as photons[1], composite bosons (e.g., bosonic atoms[2] and pairs of fermionic atoms[3]), and quasi-particles (e.g., excitons[4,5], phonons[6], polaritons[7], and magnons[8]) have been studied using a common bosonic quantum field theory. The understanding of BEC forms the foundation for exploring advanced quantum condensate states such as superfluidity and superconductivity. In particular, the behavior of bosonic particles in a periodic potential is of recent interest. Experimental, theoretical, and computational studies on both crystal lattice and optical lattice have been intensively performed[9–11].

Magnetic field-induced phase transitions in quantum magnets which approximately possess $U(1)$ global rotational symmetry are one of the experimental realizations of BEC[12–14]. The energy gap $E_g$ between the ground state and the bottom of the magnon band, which is non-zero in the quantum paramagnetic phase, is translated into chemical potential $\mu$ for the bosonic language (see inset in Fig. 1a). By tuning the applied magnetic field, $\mu$ can be tuned to be zero, resulting in a long-range ordered magnetic BEC. Unlike a cold-atom system in an optical lattice, which is another experimental realization of BEC, quantum magnets can host a much larger number of bosons, and are thus suited for the test of theoretical prediction for BEC physics in the thermodynamic limit[15].

Historically, experimental studies of BEC in quantum magnets have been mainly performed on low-spin systems such as quantum dimer systems[16–20] and $S = 1$ systems with strong easy-plane single-ion (SI) anisotropy[21]. Higher spin systems are usually considered to be disadvantageous observing quantum phenomena like BEC, while they

[1]Department of Advanced Materials Science, The University of Tokyo, Kashiwa 277-8561, Japan. [2]The Institute for Solid State Physics, The University of Tokyo, Kashiwa 277-8581, Japan. [3]Department of Engineering Science, University of Electro-Communications, Chofu 182-8585, Japan. [4]RIKEN Center for Emergent Matter Science (CEMS), Wako 351-0198, Japan. ✉e-mail: 2850487445@edu.k.u-tokyo.ac.jp; arima@k.u-tokyo.ac.jp

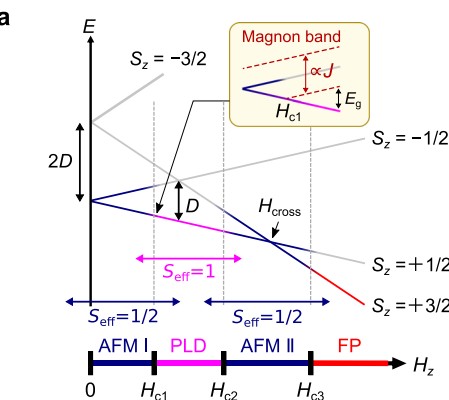

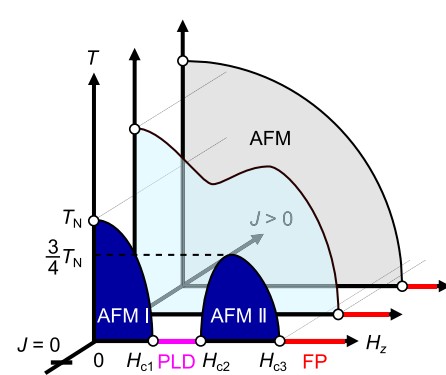

**Fig. 1 | Double Bose–Einstein-condensation (BEC) dome in $S = 3/2$ quantum magnet. a** Energy diagram of the $S = 3/2$ spin site with easy-plane type single-ion (SI) anisotropy $D\hat{S}_z^2$ ($D > 0$) in a magnetic field along the $z$-direction. The ground state mainly consists of single-site states $|S_z = \pm 1/2\rangle$ in AFMI phase, and $|S_z = 1/2\rangle$ and $|S_z = 3/2\rangle$ in AFMII phase, indicated by the dark blue colored line. Above $H_{c3}$, as well as between $H_{c1}$ and $H_{c2}$, the ground state is a direct product of the single-site ground state at that field (i.e., $|S_z = 1/2\rangle$ and $|S_z = 3/2\rangle$, respectively), indicated by the pink and red-colored lines. Antiferromagnetic interaction $J(>0)$ determines the width of magnon bands and critical fields, as schematically depicted in the inset. Each field region can effectively be described by spin Hamiltonian for $S_{eff} = 1/2$ or $S_{eff} = 1$ [see the discussion on Fig. 5d for the latter]. **b** Schematics of the phase diagram for three different strengths of $J(>0)$. In a weakly interacting regime, two BEC domes appear, separated by the intermediate PLD phase, as shown by the dark-blue color. As $J$ increases, two domes become larger and merge with each other at low temperature, as depicted by the light-blue color. Finally, in the strongly interacting regime, two domes merge entirely, indicated by the gray color.

can provide a playground for much richer physics due to the multiple excitations on a single site level.

To see the effect of the multiple excitations, let us introduce an $S = 3/2$ antiferromagnetic (AFM) Heisenberg model, given by:

$$\hat{\mathcal{H}} = J \sum_{i,j} \hat{\mathbf{S}}_i \cdot \hat{\mathbf{S}}_j + D \sum_i \left(\hat{S}_i^z\right)^2 - H_z \sum_i \hat{S}_i^z + \hat{\mathcal{H}}_{DM} + \hat{\mathcal{H}}_{me}, \quad (1)$$

where $\hat{S}_i^\alpha (\alpha = x, y, z), J(>0), D(>0), H_z, \hat{\mathcal{H}}_{DM}$ and $\hat{\mathcal{H}}_{me}$ are $S = 3/2$ spin operators at site $i$, AFM exchange interaction between neighboring sites $i$ and $j$, easy-plane SI anisotropy, magnetic field along the $z$-axis, Dzyalosinskii–Moriya (DM) interaction term and magnetoelectric coupling term, respectively. The existence of $\hat{\mathcal{H}}_{DM}$, and $\hat{\mathcal{H}}_{me}$ in $Ba_2CoGe_2O_7$ (BCGO), the material studied in this paper, allows sensitive detection of a magnetic order through electric measurements[22]. Still, as far as these two terms are small compared with the first three terms, which is the case in BCGO[23], the BEC physics would remain qualitatively unchanged from the case of the no-magnetoelectric coupling. Therefore, hereafter, we will continue the discussion without considering $\hat{\mathcal{H}}_{DM}$ and $\hat{\mathcal{H}}_{me}$. See Methods for the explicit form of the magnetoelectric coupling in BCGO.

The single-site energy diagram is depicted in Fig. 1a. In a low-field range, the system can be well described in the subspace spanned by the lower doublet $|S_z = \pm 1/2\rangle$, which yields an effective $S = 1/2$ XXZ model expressed as:

$$\hat{\mathcal{H}}_{XXZ} = J_{XXZ} \sum_{i,j} (\hat{S}_i^x \hat{S}_j^x + \hat{S}_i^y \hat{S}_j^y + \Delta \hat{S}_i^z \hat{S}_j^z) - h_z \sum_i \hat{S}_i^z, \quad (2)$$

where $J_{XXZ} = 4J$, $\Delta = 1/4$, and $h_z = H_z$[23,24]. Here, we considered the case that the ground state at a zero field is the XY-AFM order (AFM I). The application of the magnetic field splits the lower doublet into $|+1/2\rangle$ and $|-1/2\rangle$, which finally destroys the long-range order and the system enters a polarized state $\prod_i |+1/2\rangle_i$ (polarized lower doublet: PLD) at the first critical field $H_{c1}$. This magnetic-field induced phase transition in the XXZ model belongs to the universality class of BEC type[25]; in the bosonic language, $\mu$ of magnons in the PLD phase, which is proportional to $H - H_{c1}$, is positive, and magnons can no longer condense into a magnetically ordered state. With further increasing the field, the energy levels of the $|+3/2\rangle$ and $|+1/2\rangle$ meet at $H = H_{cross}$. In the vicinity of $H_{cross}$, the system can be described in the subspace

spanned by $|+3/2\rangle$ and $|+1/2\rangle$, yielding an effective $S = 1/2$ Hamiltonian Eq. (2) with $J_{XXZ} = 3J$, $\Delta = 1/3$ and $h_z = H_z - (2D + nJ)$, where $n$ is the coordination number[26]. For example, $n = 4$ for the square lattice. As a result, another antiferromagnetically ordered phase (AFM II) centered at $H_{cross}(= 2D + nJ)$ emerges between the second and third critical fields $H_{c2}$ and $H_{c3}$, which are governed by the BEC universality. At $H_{c3}$, the system eventually undergoes a phase transition to a fully polarized state (FP) $\prod_i |+3/2\rangle_i$.

The expected temperature-magnetic field phase diagram is presented in dark-blue color in Fig. 1b. Both the AFM I and AFM II phase shape dome structures called "BEC dome", and their peaks appear around $H = 0$ and $H_{cross}$, respectively. The peak temperature is proportional to $J_{XXZ}$ of the effective XXZ model in Eq. (2), i.e., $4J$ and $3J$, respectively.

Notably, the scenario mentioned above is valid only if $J$ is weak enough. As the width of each magnon band is proportional to $J$, stronger $J$ leads to an increase in $H_{c1}$ and decrease in $H_{c2}$, so the two BEC domes merge into a single phase, as shown by the light-blue region in Fig. 1b. Accordingly, an exquisite energy balance between $J$ and $D$ is required to realize this type of BEC phenomenon, which has challenged the experimental realization. In order to establish a design principle for the higher spin BEC physics, we need to investigate how the featureless single AFM phase in the strongly interacting regime evolves into two AFM domes in the weakly interacting regime as a function of $J/D$.

Here, we show the emergence of a double BEC dome in well-known multiferroic material $Ba_2CoGe_2O_7$ (BCGO)[22], where $J$ is effectively tuned by diluting the magnetic Co sites with Zn. We observed a systematic reduction of the transition temperature as a function of Zn concentration and confirmed the phase diagram with the double BEC dome structure by measurements using a pulsed high magnetic field and low-temperature experiments. Mean-field calculations excellently reproduced the observations except for the quantum critical point between the first BEC dome and PLD, which can be explained by a theory for the dirty bosons.

## Results

### Magnetic properties of non-doped BCGO

BCGO is a quasi-two-dimensional magnet comprised of a simple square lattice of magnetic $Co^{2+}(3d^7)$ ions surrounded by four $O^{2-}$ ions [Fig. 2a, b]. Each $CoO_4$ tetrahedron is strongly compressed along the

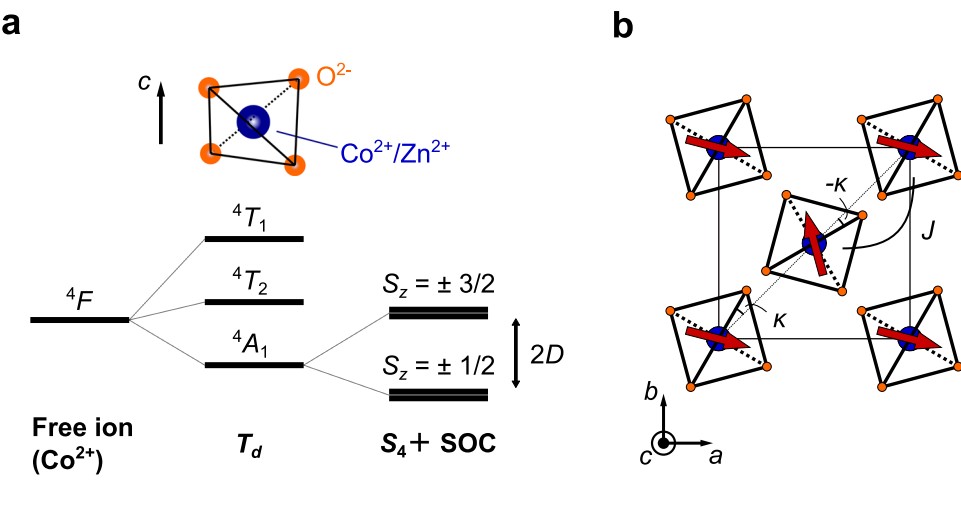

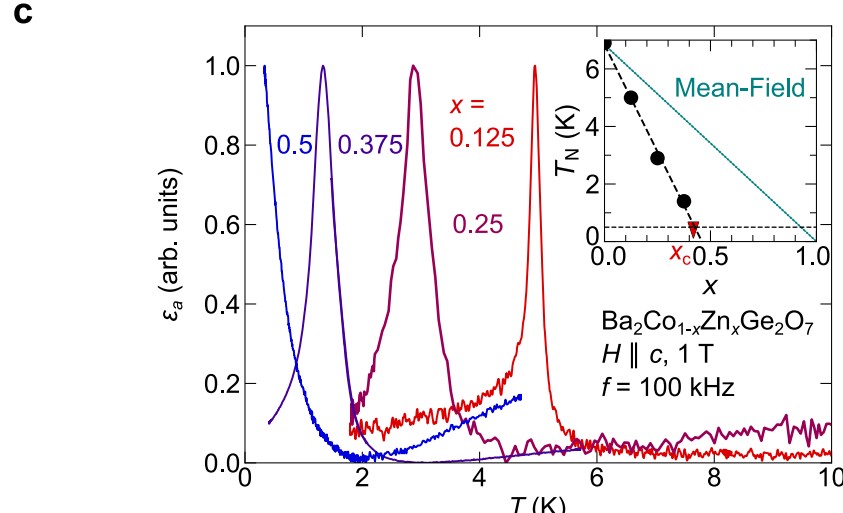

**Fig. 2 | Basic physical properties of Ba$_2$Co$_{1-x}$Zn$_x$Ge$_2$O$_7$ (BCZGO). a** Energy level splitting of $3d$ electrons at Co$^{2+}$. The free-ion state splits into three levels in the $T_d$ crystal field environment. The lowest four-fold degenerate $^4A_1$ state can be described by $S = 3/2$ with easy-plane anisotropy ($D\hat{S}_z^2$) due to the $S_4$ crystal field and spin-orbit coupling (SOC). **b** Schematic magnetic structure of Ba$_2$CoGe$_2$O$_7$ (BCGO). Red arrows represent spin orientations in the ordered phase. The spin canting is exaggerated from the actual angle for visibility (see Supplementary Note 1).

**c** Temperature dependence of electric permittivity $\varepsilon_a$ along the $a$-axis of BCZGO. Zero levels are also shifted. Inset shows $x$ dependence of antiferromagnetic transition temperature $T_N$. The geometrical percolation threshold $x_c = 0.41$ of the square-lattice is shown as a red triangle. In $x = 0.5$, no sign of long-range order was observed down to 0.4 K. The mean-field line of $T_N(x)/T_N(0) = 1 - x$ is shown by a green dotted line.

c-axis, which reduces the local symmetry at the Co$^{2+}$ site from $T_d$ to $S_4$ [Fig. 2a]. The $S_4$ crystal-field and spin-orbit coupling induce strong easy-plane SI anisotropy $D \approx 1.03$ meV compared with the AFM exchange interaction $J \approx 0.21$ meV[23,27,28]. AFM long-range order appears below $T_N = 6.7$ K at zero field, and the magnetic phase diagram of BCGO consists of one broad AFM phase, indicating that this material is in the strongly-interacting regime [Fig. 1b]. Therefore, BCGO is ideal for observing the transition from the strongly interacting regime to the weakly interacting regime. BCGO is known to host the multiferroic property[29–31], which is well explained by the spin-dependent $p$-$d$ hybridization mechanism[32]. Due to the strong magnetoelectric coupling in the ordered phase, the dielectric measurement enables us to sensitively detect magnetic phase transition as well as to explore the critical phenomena[22].

### Reduction in the effective magnetic interaction
First, we demonstrate a systematic reduction in the effective magnetic interaction $J_{eff}$ from the original $J \approx 0.21$ meV by the partial substitution

of non-magnetic Zn$^{2+}$ ions. Figure 2c shows the temperature dependence of relative electric permittivity $\varepsilon_a$ along the $a$-axis in Ba$_2$Co$_{1-x}$Zn$_x$Ge$_2$O$_7$ (BCZGO, $x = 0.125$, 0.25, 0.375, and 0.5). The electric permittivity exhibits a peak at the transition temperature, except for $x = 0.5$, where no obvious sign of long-range order is observed down to 0.4 K. As $x$ increases, the ordering temperature decreases linearly, as observed in other two-dimensional magnets with rotational symmetry[33]. The extrapolation of $T_N$ reaches the axis of abscissas at $x = 0.45$ [inset of Fig. 2c], which is close to the percolation threshold of the square lattice: $x_c = 0.41$[34]. Beyond $x_c$, the system is composed of finite-size magnetic clusters, and long-range magnetic order cannot arise in definition, which is consistent with the absence of the peak in $\varepsilon$ down to 0.4 K for $x = 0.5$.

### Magnetic field dependences of physical properties
Next, Fig. 3 (a) presents the magnetization curves of BCZGO ($x = 0.25$, $T_N = 2.8$ K) for $H\|c$ measured in pulsed high magnetic fields of up to 50 T at 1.4 and 4.2 K. At 1.4 K, the magnetization $M$ monotonically

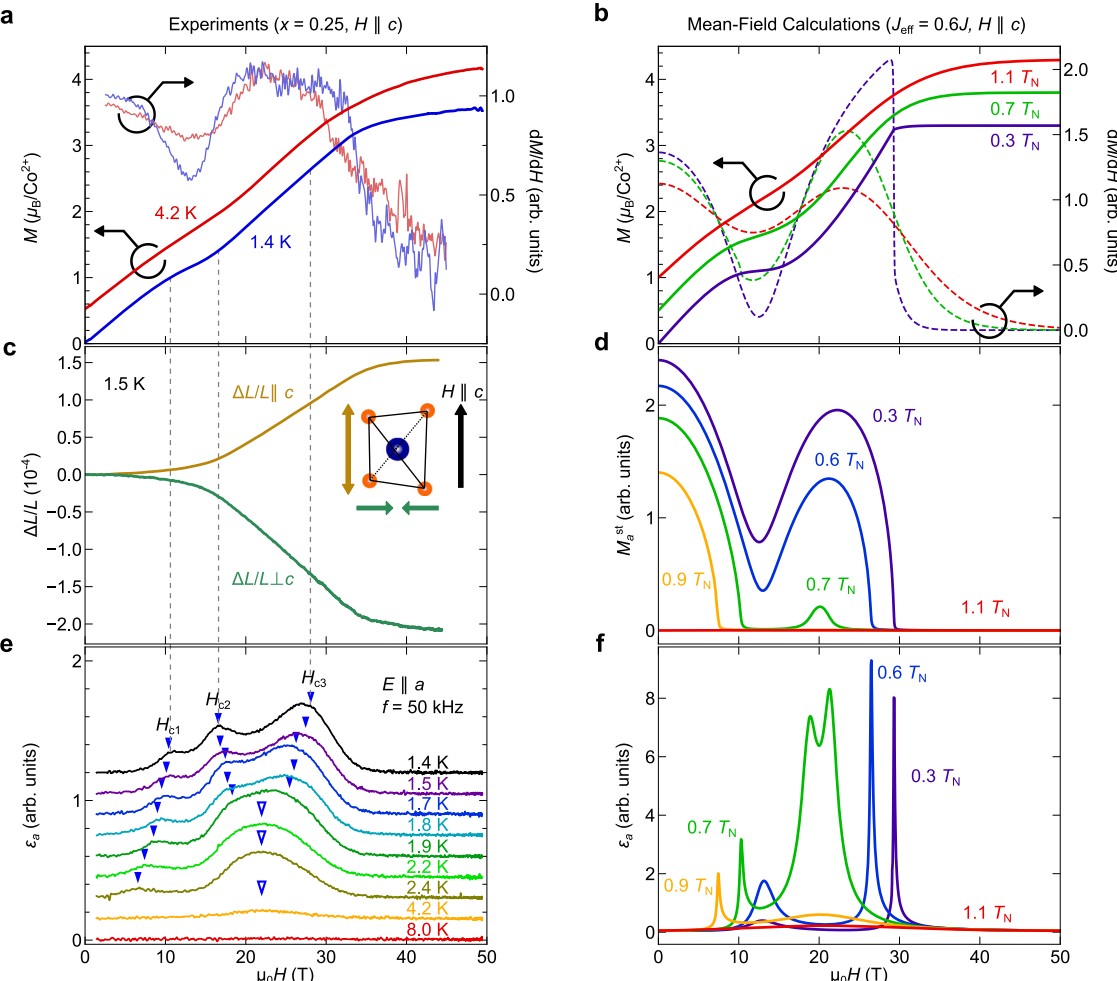

**Fig. 3 | Magnetic field dependence of physical properties of BCZGO. a** Magnetic-field dependence of magnetization $M$ and its field derivatives $dM/dH$. Each $M$–$H$ curve is offset vertically for clarity. **c** Magnetostriction, measured for the striction along the $c$-axis (yellow, $\Delta L/L \| c$) and perpendicular to the $c$-axis (green, $\Delta L/L \perp c$). **e** Magnetic field dependence of $\varepsilon_a$ at various temperatures. Each $\varepsilon$–$H$ curve is offset vertically for clarity. Phase boundaries are defined as blue-closed triangles

determined from the field derivative of $\varepsilon_a$ (see Supplementary Note 2). Broad peaks around 22 T at higher temperatures are marked by the blue-open triangles. **b**, **d**, and **f** Calculated field dependences of $M$, staggered magnetization $M_a^{st}$ of $a$-axis component, and $\varepsilon_a$. Temperatures are normalized by the $T_N(H = 0)$. The scale factor for the interaction term $J$ is set to 0.6. See "Methods" for the detail.

increases until the saturated value of $M_{sat} \approx 3.3~\mu_B/Co^{2+}$ is reached at ~ 35 T, except for the plateau-like anomaly around 12 T approximately at one-third of $M_{sat}$, indicating the emergence of the PLD phase. The anomaly broadens but remains visible at 4.2 K. These features were well reproduced by the mean-field calculation, for which the same Hamiltonian with Ref. [24] was adopted, and the dilution effects were effectively incorporated by tuning the exchange interaction $J_{eff}$ [Fig. 3b] (see "Methods"). Figure 3c shows the magnetostriction [$(\Delta L/L)_\parallel$ and $(\Delta L/L)_\perp$] parallel and perpendicular to the $c$-axis in $H \| c$ at 1.5 K. No obvious magnetostrictive anomaly can be seen at the onset of the magnetization plateau, suggesting that the exchange striction in BCZGO is weak partly due to the long exchange path between $Co^{2+}$ sites [Fig. 2b]. Meanwhile, a rapid increase in $(\Delta L/L)_\parallel$ and a decrease in $(\Delta L/L)_\perp$ with magnitudes of ~$10^{-4}$ were observed in response to the further increase in $M$ between 15 and 35 T. This should be attributed to the crystal-field striction; the contribution of $|+3/2\rangle$ to the ground state should elongate the $CoO_4$ octahedra along the $c$-axis. This type of magnetostriction is distinct from that previously reported in BEC materials such as $NiCl_2 \cdot 4SC(NH_2)_2$[35], where $\Delta L/L$ changes nonmonotonically during passage through the BEC dome due to the exchange striction mechanism. Note that the saturation field observed in $M$ and $\Delta L/L$

measurements is expected to be close to the critical field to the FP phase at absolute zero[17,36], but does not agree quantitatively due to the finite temperature phase boundary, as indicated by the calculation of staggered magnetization $M_a^{st}$ [Fig. 3d], which is the order parameter of the XY-AFM order.

Established the magnetic properties of BCZGO, we examine magnetic phase transitions by dielectric constant measurement. Figure 3e shows a magnetic-field dependence of $\varepsilon_a$ at various temperatures, simultaneously measured with $M$ shown in Fig. 3a. At 1.4 K, three peaks are observed at $H_{c1} = 11$ T, $H_{c2} = 16$ T, and $H_{c3} = 26$ T, excellently agreeing with the prediction for phase transitions across AFMI → PLD → AFM II → FP phase transitions. As the temperature increases from 1.4 K, $H_{c1}$ and $H_{c3}$ shift to the lower field; whereas, $H_{c2}$ shifts to the higher field. These dielectric responses were reproduced by the mean-field calculation [Fig. 3f]. Moreover, calculated $M_a^{st}$ value [Fig. 3d] also supports that the three peaks in $\varepsilon_a$–$H$ corresponds to the phase boundaries.

**Magnetic phase diagram of BCZGO**
Based on the $\varepsilon_a$ measurements, we construct the magnetic phase diagram for BCZGO with $x = 0.25$, as shown in Fig. 4. In contrast to undoped BCGO, the magnetic phase diagram is characterized by a

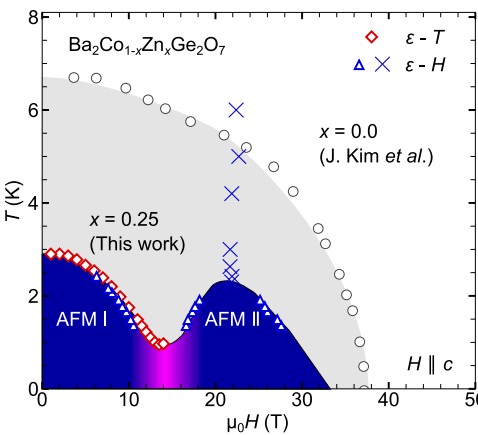

**Fig. 4 | Magnetic phase diagrams of BCZGO.** The points represented by blue markers are determined by the $\varepsilon$–$H$ measurements in a pulsed magnetic-field [Fig. 3e], while those shown by red markers are obtained from the $\varepsilon$–$T$ measurements in a persistent magnetic-field data shown in Fig. 5a. The black open circles correspond to the phase boundary of pristine Ba$_2$CoGe$_2$O$_7$, which are collected from ref. [22].

double-dome structure. The peak temperature of the second BEC dome is estimated to be 2.1 K, which is consistent with the prediction $(3/4)T_N$ [Fig. 1b], and the two domes merge below 1 K. The blue crosses at around 22 T in Fig. 4, which is observed as a broad peak in $\varepsilon$–$H$ [Fig. 3e], are not phase boundary. They instead correspond to the enhancement of the spin fluctuation due to the level-crossing at the $H_{cross}$ in Fig. 1a. Indeed, considering the effective $S = 1/2$ XXZ model Eq. (2), the second BEC dome is expected to take its maximum around $H_{cross} = 2D + nJ$. We obtain a consistent estimation of $H_{cross}$ to be 22.4(4) T for BCZGO ($x = 0.25$), where we use the reported values of $J_z$, $D$, $g_{zz}$, where $g_{zz}$ is the $z$-component of $g$-factor, for BCGO [23,28] and assuming a reduction from $n = 4$ to $n = 4(1-x)$ due to the dilution.

In $x = 0.25$, the PLD phase only exists at finite-temperature. Further reduction of effective magnetic interaction by the dilution would be required to separate two BEC domes completely. In this case, however, we need to carefully consider the effect of randomness, which brings another possibly that the site dilution may lead to the trivial percolation disorder instead of the proposed three distinct BEC phase transitions in Fig. 1.

To confirm the relevance of the randomness effect, we measured $\varepsilon_a$ for higher doped sample $x = 0.375$ at the lower temperatures down to 0.4 K in static magnetic fields. Figure 5a, b summarize a magnetic-field dependence of the ordering temperature $T_N(H)$ for $x = 0.25$ and 0.375, respectively, which are extracted from the peak positions of the $\varepsilon$–$T$ curves [insets of the figures]. By fitting $T_N(H)$ using a power-law scaling $T_N(H) \propto |H - H_c|^{\varphi}$, the critical exponent $\varphi$ estimated to be $\varphi = 1.07(1)$ for $x = 0.375$ [red line in Fig. 5 (b)]. The phase diagram for $x = 0.25$ also has a region where the fitting yields $\varphi = 0.93(2)$, close to 1 [red line in Fig. 5a]. We cannot reconcile these results with the theoretical predictions for conventional systems such as $\varphi = 2/3$ for three-dimensional BEC systems or $\varphi = 1/2$ for three-dimensional Ising systems [37]. We note, however, $\varphi \approx 1$ was reported for the systems with quenched disorder both in experiments and a numerical simulation [36,38,39]. The phase transition characterized by $\varphi \approx 1$ was attributed to a transition from inhomogeneous BEC to Bose glass (BG), i.e., local ordering (BEC) clusters weakly connect to show a long-range ordered phase at the first state; they are eventually disconnected, leaving only locally ordered clusters. Therefore, $\varphi \approx 1$ in BCZGO implies that the field-induced disordred phase is the BG phase [Fig. 5b]. These results show that the randomness effect plays a central role in this

system. We discuss a model on this line to capture the phase transition features in BCZGO as follows.

## Discussion

Here, we theoretically discuss the possibility of complete separation of two BEC domes by explicitly considering the randomness effect induced by the site dilution. First, in the vicinity of $H_z \approx (1/2)H_{cross}$, the system can be described in a subspace spanned by $\{|3/2\rangle, |1/2\rangle,$ and $|-1/2\rangle\}$ [Fig. 1a], which we can translate into the Hilbert space expressed by the operators for $S = 1$. $S = 3/2$ spin operators effectively take the form,

$$
\begin{aligned}
(\hat{S}_x)_{eff} &= \frac{1}{\sqrt{2}}\left(1 + \frac{\sqrt{3}}{2}\right)\hat{S}_x + \frac{1}{\sqrt{2}}\left(-1 + \frac{\sqrt{3}}{2}\right)\hat{Q}_{zx}, \\
(\hat{S}_y)_{eff} &= \frac{1}{\sqrt{2}}\left(1 + \frac{\sqrt{3}}{2}\right)\hat{S}_y + \frac{1}{\sqrt{2}}\left(-1 + \frac{\sqrt{3}}{2}\right)\hat{Q}_{yz}, \\
(\hat{S}_z)_{eff} &= \hat{S}_z + \frac{1}{2}I,
\end{aligned}
\tag{3}
$$

where $I$ is an identity operator and quadrupole operators are defined as $\hat{Q}_{zx} \equiv \hat{S}_z\hat{S}_x + \hat{S}_x\hat{S}_z$ and $\hat{Q}_{yz} \equiv \hat{S}_y\hat{S}_z + \hat{S}_z\hat{S}_y$. Omitting the quadrupole terms, as its coefficients are more than ten times smaller than usual spin operators, the effective Hamiltonian becomes equivalent to the $S = 1$ XXZ model with SI anisotropy

$$
\begin{aligned}
\hat{\mathcal{H}}_{XXZ} = {}& J_{XXZ}\sum_{ij}\left(\hat{S}_i^x\hat{S}_j^x + \hat{S}_i^y\hat{S}_j^y + \Delta\hat{S}_i^z\hat{S}_j^z\right) \\
& + D\sum_i\left(\hat{S}_i^z\right)^2 - h_z\sum_i\hat{S}_i^z,
\end{aligned}
\tag{4}
$$

where $J_{XXZ} \approx 1.74J$, $\Delta \approx 0.57$, and $h_z = H_z - (D + nJ/2)$ [see Fig. 5c]. The effective magnetic field $h_z$ stabilizes the XY magnetic order. Therefore, the ordering temperature, if exists, should take a minimum around $h_z = 0$, i.e., $H_z = D + nJ/2 = (1/2)H_{cross}$. Note that an actual $H_z$ at which the system would have the lowest ordering temperature is $(1/2)H_{cross} < H_z < (4/7)H_{cross}$ due to the quadrupole terms. See Supplementary Note 3 for the detail.

Site-dilution effect in the model Eq. (4) has been, in fact, investigated theoretically in ref. [40], where the exchange anisotropy $\Delta$ was fixed to 1. Note that $\Delta$ does not have a significant effect in the present context because the possible ordered phase does not have any $z$-component even at zero field due to the SI anisotropy $D$ [41]. The resultant simplified phase diagrams as a function of $x$ for representative $D/J_{XXZ}$ are shown in Fig. 5d. The ground state at a zero field ($h_z = 0$) in a clean limit ($x = 0$) is the XY ordered state for $D/J_{XXZ} < 5.5$ and the quantum paramagnetic state for $D/J_{XXZ} > 5.5$. For $2.3 \lesssim D/J_{XXZ} \lesssim 5.5$, the theory predicts the emergence of a disordered phase so-called Mott glass (MG) phase in a region $x^* < x < x_c$. This phase is a dilution-induced non-geometrically percolating quantum disordered phase, and corresponds to the PLD phase in the current context. Using the reported parameters, we ensure that BCGO is in this parameter regime as $D/J_{XXZ} \approx 3.0$ [23,28], which theoretically corroborates that the complete separation of two BEC domes would be possible in BCZGO.

Experimental results also support the above discussion. The effective zero-field point ($h_z = 0$) of the $S = 1$ model is estimated to be $H_z = 11.2$ T and 11.0 T for $x = 0.25$ and 0.375, respectively (see Supplementary Note 4). For $x = 0.25$, we observe magnetic order at the estimated $h_z = 0$ ($H_z = 11.2$ T), consistent with the XY ordered ground state in the phase diagram for $D/J_{XXZ} \approx 3.0$ in ref. [40]. It is hence expected that $0.25 < x^*$. However, for $x = 0.375$, the experimentally obtained $H_{c1}$ is 11.3(3) T, close to the predicted $h_z = 0$ ($H_z = 11.0$ T). This is consistent with the theoretical prediction that $x^*$ is slightly smaller than 0.375.

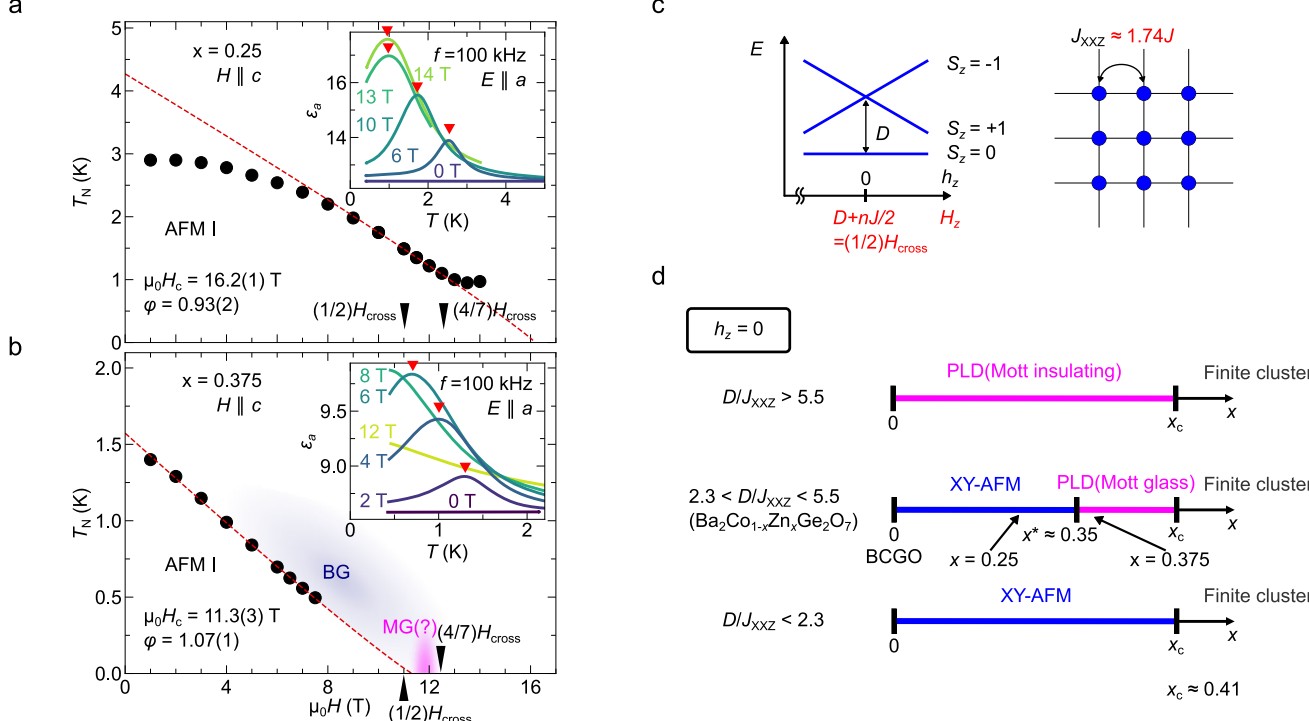

**Fig. 5 | Beyond mean-field approach and randomness effect. a, b** Low-temperature phase diagram for $x = 0.25$ and $0.375$, respectively. The latter contains provisional suggestions on Bose-glass (BG) and Mott-glass (MG). The fittings to the power-law $T_N \propto |H - H_c|^{\varphi}$, are also shown by red dashed lines. The predicted values of $(1/2)H_{cross}$ and $(4/7)H_{cross}$ from the previous reports[23,28] are shown as black triangles (see text). Inset shows the temperature dependence of $\varepsilon_a$, which is used to obtain the ordering temperatures (red triangles). **c** Schematics of effective $S = 1$

XXZ model with SI easy-plane anisotropy $D\hat{S}_z^2$ on a square-lattice. Original multiplet, $S_z = 3/2, 1/2, -1/2$ for $S = 3/2$ are mapped on to $S = 1$, $S_z = 1, 0, -1$ states [see Fig 1a and text]. Magnetic field $h_z$ and magnetic interaction $J_{XXZ}$ in $S = 1$ model corresponds to $D + nJ/2$ and $1.74J$ in original $S = 3/2$ model, respectively. **d** Ground state magnetic phase diagram as a function of $x$ for diluted $S = 1$ XXZ model at $h_z = 0$. Line for each $D/J_{XXZ}$ is reproduced from Fig. 1 in ref. [40]. $\overset{*}{x} \approx 0.35$ is the critical value for $D/J_{XXZ} \approx 3$, which corresponds to BCGO.

The possible quantum disordered phase around 11 T would correspond to the MG phase [Fig. 5b]. Therefore, we may provide BCZGO, unlike other diluted systems[42,43], as a candidate stage for the dilution-induced quantum phase transition in an insulator spin system without crossing the percolation threshold.

To summarize, we have investigated the evolution of the magnetic phase diagram of BCZGO and confirmed that the effective magnetic interaction is tunable via the dilution approach. In $x = 0.25$, due to the effective reduction of the interaction, the magnetic field dependence of electric permittivity shows a three-peak structure, which suggests the emergence of a double BEC-dome. The randomness effect, which is beyond the mean-field treatment of dilution, is also observed in a comprehensible way. In $x = 0.375$, the ordering temperatures of the first BEC dome obey the scaling law with $\varphi \approx 1$, which belongs to the universality of the BEC-Bose glass phase transition. Thus, the resulting paramagnetic phase can be regarded as a Bose glass phase. Effective Hamiltonians at the different field regions well explain the experimentally obtained magnetic phase diagram of BCZGO.

Owing to the simpleness of each effective model, BCZGO would be an excellent playground for theoretical and numerical studies of the bosonic particle systems with quenched disorder. We note that the discussion used here can be straightforwardly generalized to higher spin ($S > 1$) systems with strong easy-plane anisotropy. For example, an $S = 2$ system can be well described in a subspace spanned by $|S_z = 0\rangle$ and $|S_z = \pm1\rangle$ at a zero field. This is actually confirmed in $Ba_2FeSi_2O_7(S = 2)$[44–47]. Particularly in ref. [44], they also derived effective $S = 1$ Hamiltonian in a zero field and yield $D/J_{XXZ} \approx 5.4$, which is close to the critical point of $D/J_{XXZ} \approx 5.5$. Therefore, a small amount of dilution could induce non-geometrically percolating quantum phase transition. This general criterion may give rise to the further discovery of

BEC materials and, consequently, advanced understanding of magnetic BEC phenomena.

## Methods
### Samples
Single crystals of $Ba_2Co_{1-x}Zn_xGe_2O_7$ ($x = 0, 0.125, 0.25, 0.375, 0.5$) were grown by floating zone method in 3 atm of $O_2$ atmosphere. $BaCO_3$, $CoO$, $ZnO$, and $GeO_2$ powders were mixed in the stoichiometric ratio and calcined at 980 °C for 60 h with multiple intermediate grinds. The resultant powder was reground, pressed into a rod, and sintered at 950 °C for 24 h in air to form seed and feed rods. Crystal axes were determined by back-reflection Laue X-ray photographs.

### Measurements of electric permittivity, magnetization, and magnetostriction
Electric permittivity $\varepsilon$ up to 9 T above 1.8 K was measured using an LCR meter (E4980A/B, Agilent) in a commercial cryostat equipped with a superconducting magnet (PPMS, Quantum Design). Two electrodes were attached to the large (100) surfaces of the polished thin plate crystals. To achieve low temperature down to 0.4 K in static fields, ${}^3$He refrigerator (Heliox, Oxford Instruments) was used, which was inserted into the cryostat equipped with a 12 T/14 T superconducting magnet. Magnetization $M$ below 7 T was measured using a SQUID magnetometer (MPMS-XL, Quantum Design). The higher-field $M$ and $\varepsilon$ up to 50 T were simultaneously measured using a non-destructive pulsed magnet (36 ms duration) at the Institute for Solid State Physics (ISSP). $M$ was measured by the conventional induction method using coaxial pickup coils. Capacitance was measured along the electric field direction ($E\|a$) by using a capacitance bridge (General Radio 1615-A) and converted to $\varepsilon$[48]. Magnetostriction $\Delta L/L$ up to 44 T was measured

by the optical fiber-Bragg-grating technique using the optical filter method in a non-destructive pulsed magnet (36 ms duration) at ISSP[49].

## Mean field calculations

Numerical simulation used in this study is based on the Hamiltonian proposed in[24]. Using $S = 3/2$ operators, the Hamiltonian of BCGO can be written as:

$$
\begin{aligned}
\hat{\mathcal{H}} = &\, J \sum_{(i,j)} \left( \hat{S}_i^x \hat{S}_j^x + \hat{S}_i^y \hat{S}_j^y \right) + J_z \sum_{(i,j)} \hat{S}_i^z \hat{S}_j^z \\
&+ D \sum_i \left( \hat{S}_i^z \right)^2 - g\mu_B \mathbf{H} \cdot \sum_i \hat{\mathbf{S}}_i - \mathbf{E} \cdot \sum_i \hat{\mathbf{P}}_i \\
&+ J_{DM} \sum_{i \in A, j \in B} \left( \hat{S}_i^x \hat{S}_j^y - \hat{S}_i^y \hat{S}_j^x \right) + K_z \sum_{(i,j)} \hat{P}_i^z \hat{P}_j^z,
\end{aligned}
\tag{5}
$$

where $J_{DM}$, $K_z$, and $\mathbf{E}$ are Dzyalosinskii-Moriya interaction and spin nematic interaction, and electric field, respectively. DM term corresponds to $\hat{\mathcal{H}}_{DM}$ in Eq. (1), and the fifth and seventh terms correspond to $\hat{\mathcal{H}}_{me}$ in Eq. (1). The cartesian cordinate $x$, $y$, and $z$ correspond to the crystallographic [110], [$\bar{1}$10], [001] directions, respectively. $\hat{P}_i^\alpha (\alpha = x, y, z)$ is the spin-induced electric polarization at site $i$, calculated using:

$$
\begin{aligned}
\hat{P}_i^x &\propto -\cos 2\kappa_i \left( \hat{S}_i^x \hat{S}_i^z + \hat{S}_i^z \hat{S}_i^x \right) - \sin 2\kappa_i \left( \hat{S}_i^y \hat{S}_i^z + \hat{S}_i^z \hat{S}_i^y \right), \\
\hat{P}_i^y &\propto \cos 2\kappa_i \left( \hat{S}_i^y \hat{S}_i^z + \hat{S}_i^z \hat{S}_i^y \right) - \sin 2\kappa_i \left( \hat{S}_i^x \hat{S}_i^z + \hat{S}_i^z \hat{S}_i^x \right), \\
\hat{P}_i^z &\propto \cos 2\kappa_i \left[ \left( \hat{S}_i^y \right)^2 - \left( \hat{S}_i^x \right)^2 \right] - \sin 2\kappa_i \left( \hat{S}_i^x \hat{S}_i^y + \hat{S}_i^y \hat{S}_i^x \right),
\end{aligned}
\tag{6}
$$

where $\kappa_i$ is a rotation angle of $CoO_4$ tetrahedron around the site around the $c$-axis; $\kappa_{i \in A} = \kappa$, $\kappa_{i \in B} = -\kappa$ [see Fig. 2b]. The mean field Hamiltonian is given by:

$$
\hat{\mathcal{H}}_{MF} = \hat{\mathcal{H}}_A + \hat{\mathcal{H}}_B,
\tag{7}
$$

where $\hat{\mathcal{H}}_A$ is

$$
\begin{aligned}
\hat{\mathcal{H}}_A = &\, 4J\hat{\mathbf{S}}_A \cdot \langle \hat{\mathbf{S}}_B \rangle + D\left( \hat{S}_A^z \right)^2 - g\mu_B \mathbf{H} \cdot \hat{\mathbf{S}}_A - \mathbf{E} \cdot \hat{\mathbf{P}}_A \\
&+ 4J_{DM} \left( \hat{S}_A^x \langle \hat{S}_B^y \rangle - \hat{S}_A^y \langle \hat{S}_B^x \rangle \right) + 4K_z \hat{P}_A^z \langle \hat{P}_B^z \rangle,
\end{aligned}
\tag{8}
$$

The mean values of $\langle \hat{\mathbf{S}}_B \rangle$ and $\langle \hat{\mathbf{P}}_B \rangle$ are calculated as functions of $\langle \hat{\mathbf{S}}_A \rangle$ and $\langle \hat{\mathbf{P}}_A \rangle$ as:

$$
\langle \hat{O}_B \rangle = \frac{\mathrm{Tr}(\hat{O}_B e^{-\beta \hat{H}_B(\langle \hat{\mathbf{S}}_A \rangle, \langle \hat{\mathbf{P}}_A \rangle)})}{\mathrm{Tr} e^{-\beta \hat{H}_B(\langle \hat{\mathbf{S}}_A \rangle, \langle \hat{\mathbf{P}}_A \rangle)}},
\tag{9}
$$

where $\langle \hat{O} \rangle_B$ is a general operator representing either $\langle \hat{S} \rangle_B$ or $\langle \hat{P} \rangle_B$. $\hat{H}_B$ is obtained by exchanging the sublattice $A$ and $B$. Here, we assume $J = J_z$. Using a scale factor $\gamma$ ($0 < \gamma < 1$), reduced effective interactions can be expressed as $\gamma J$, $\gamma J_{DM}$, $\gamma K$, respectively. To lift the degeneracy, small electric field was applied in the $a$-axis. Electric permittivity can be calculated using:

$$
\varepsilon_a \equiv \frac{\langle \hat{P}_a(E_1) \rangle - \langle \hat{P}_a(E_2) \rangle}{(E_1 - E_2)}
\tag{10}
$$

with small applied electric field $E_1$ and $E_2$ in the $a$-axis.

## Data availability

All data that support the findings of this study are included in this published article and supplementary information files. Source data are provided as a Source Data file. Source data are provided with this paper.

## Code availability

The code used for this study is provided as a Supplementary Software.

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

## Acknowledgements

This work was partly supported by JSPS Grants No. 19H01835, No. JP19H05824, and No. JP19H05826, and by CREST (No. JPMJCR19T5) from Japan Science and Technology (JST). Y.W. are supported by JSPS through Program for Leading Graduate Schools (MERIT). Measurements of electric permittivity above 1.8 K, X-ray Laue photograph, high-field magnetostriction, high-field magnetization, and high-field electric permittivity were performed at the Institute for Solid State Physics (ISSP), the University of Tokyo, Japan.

## Author contributions

Y.W., Y.T., and T.A. conceived the project. Y.W. grew single crystals. Electric permittivity measurements above 1.8 K in static magnetic fields were performed by Y.W. High-field magnetostriction measurements were performed by M.G. and Y.W. under the supervision of A.I. High-field magnetization and electric permittivity measurements were performed by A.M. and Y.W. under the supervision of M.T. Y.W. constructed the theoretical model and performed numerical calculations. Y.W., Y.M., and K.H. designed the electric permittivity measurement setup and performed the low-temperature measurements using ³He refrigerator under the supervision of T.S. All authors discussed the results. Y.W. and M.G. wrote the manuscript with the inputs from all authors. T.K., Y.T., and T.A. supervised the projects.

## Competing interests

The authors declare no competing interests.
