## [Peer Review File · Nature Communications]

REVIEWER COMMENTS

Reviewer #1 (Remarks to the Author):

The manuscript "Bose-Einstein Condensations in Diluted $S = 3/2$ Quantum Magnets: Evolution from single- to double-dome structure" by Watanabe and coworkers reports on the properties of a BEC system with a spin of $3/2$. The probe used to explore the BEC phase diagram is the level of magnetic dilution in $\text{Ba}_2\text{CoGe}_2\text{O}_7$ where Co is partially exchanged with Zn leading to theoretically predictable changes. The authors find in addition, that magnetic dilution is leading to quenched disorder and percolation in $\text{Ba}_2\text{Co}_{1-x}\text{Zn}_x\text{Ge}_2\text{O}_7$ as a result of the formation of magnetic clusters thereby suppressing long-range magnetic order.

The authors went to great length to measure the temperature-magnetic field phase diagram of $\text{Ba}_2\text{Co}_{1-x}\text{Zn}_x\text{Ge}_2\text{O}_7$ single crystals ($x=0, 0.125, 0.25, 0.375, 0.5$) using magnetostriction and electric permittivity in magnetic fields up to 50T. Their reported findings are supported and confirmed by theoretical model calculations. It is also interesting to note that the theoretical discussion can be extended to spin systems with $S>2$ and strong easy-plane anisotropy. This will certainly help to gain a good understanding of the magnetic properties of BEC systems in general.

Overall the manuscript is well presented, the experiment and their findings explained clearly and the results are consistent in particular that the theoretical modelling seems to work well. I can therefore recommend the manuscript for publication in NCOMMS.

Reviewer #2 (Remarks to the Author):

General comments

This is an interesting paper although I am not convinced it meets the criterion of quality and originality expected for Nature Communications.

The authors explore the effects of magnetic field on a spin- $3/2$ antiferromagnet with weak planar anisotropy. They use doping as a way to control the exchange interactions and observe 2 connected domes of Bose-Einstein condensates as a function of field. Of course, an order state with gapless

excitation is a trivial example of a Bose-Einstein condensate which is routinely measured. However it is possible to use a magnetic field to tune a system out of this phase by making the excitations gapped. In the case of spin >1 and planar anisotropy which separate the spin components in energy, it is possible to enter further BEC phases as the higher spin states can condense into the ground state with increasing field.

The authors show this process in a qualitative way. They use a spin-3/2 magnet which orders magnetically and has a single BEC dome due to the strong exchange interactions between the magnetic ions. By doping they are able to control the exchange interactions allowing them to explore the weak interaction regime where 2 domes of Bose-Einstein condensates are observed as a function of field.

I call their results qualitative because strictly speaking one needs to confirm the correct critical exponent to prove BEC whereas their exponents are substantially different probably due to disorder effects from doping which they indeed discuss. Furthermore they do not actually see the expected 2 domes, rather their BEC domes are joined. Greater doping might separate the domes better as a function of field but would introduce more disorder, hence this is not further explored in detail.

Altogether the paper presents an interesting idea but it would be better to observe it in a clean undoped system with weaker interactions, where the 2 BEC domes are naturally separated and can be explored quantitatively. It would be good if the authors gave a full justification of their reasoning behind the choice of material for this study.

Other comments

Page 3

'The anomaly becomes blunt' -> 'The anomaly becomes broadened'

'suggesting the weak exchange striction in BCZGO partly due to the long exchange path' -> 'suggesting that the exchange striction in BCZGO is weak partly due to the long exchange path'

'but does not in accordance with the finite temperature phase boundary' -> 'but does not agree quantitatively due to the finite temperature phase boundary'

Fig 2

It would be clearer if parts a) to c) which describe the basic structural properties were in a different figure from parts d) to i)

Reviewer #3 (Remarks to the Author):

The authors study Bose-Einstein condensation (BEC) in the compound series $\text{Ba}_{1-x}\text{Zn}_x\text{Ge}_2\text{O}_7$. The essential physics is captured by a relatively simple model of $S=3/2$ spins on the square lattice with easy-plane single ion anisotropy D , interacting via the Heisenberg exchange interaction. When D is small compared to the exchange coupling J , the antiferromagnetic region in the longitudinal magnetic field vs. temperature phase diagram is featureless. However, when J is lowered a double-peak structure of the same phase appears. In the limiting case of very small J the double peak structure actually evolves into two separate domes that can be seen as two different Bose-Einstein condensates. This structure is richer than the single BEC dome that appears in most of the previous theoretical and experimental works, though it could be somewhat reminiscent of that of spin-1 couple dimer compounds (see $\text{Ba}_3\text{Mn}_2\text{O}_8$).

The Hamiltonian that is supposed to describe $\text{Ba}_2\text{CoGe}_2\text{O}_7$ more accurately is actually more complicated and, in particular, includes a certain form of magnetoelectric coupling; the latter is what allows to use the measurements of the electric permittivity to detect the magnetic phase transitions (more conventional magnetic measurements are also performed). The big theoretical assumption is that the same kind of Hamiltonian describes the chemically substituted compounds, with the only difference that the ratio J/D becomes smaller; in other words, the dilution of the magnetic sites is taken into account phenomenologically by decreasing J in the same Hamiltonian. This proves to be a reasonable assumption by looking at the fair agreement between the experimental data and the mean-field calculation based on such theoretical model. As expected, there are features that cannot be explained within this framework, such as the critical exponents (related to the shape of the phase transition line) near the quantum critical point; in principle it should be possible to do a more systematic analysis of the effect of disorder to deal with this aspect, but it is fair to say that it goes beyond the scope of this paper and the authors give sensible arguments based on the existing literature.

I would like the authors to address the following points.

i) By looking at the various figures, I am under the impression that the authors have enough data to draw the H-T phase diagram also for $x=0.375$. If this is true, it would be nice to show that phase diagram as well. Even if the nature of the non-magnetic phase at low T remains somewhat elusive as discussed in the text, that could be indicated in the caption and/or the figure.

ii) The authors should make a little clearer that, while they use Eq.(1) as a pedagogical model to explain the essential physics, as soon as they start to deal more concretely with $\text{Ba}_{1-x}\text{Zn}_x\text{Ge}_2\text{O}_7$ they actually use Eq.(5). I think that would actually make it easier for the reader to immediately appreciate the rationale and the value of the permittivity measurements.

iii) I would recommend to thoroughly check the English; e.g., "Therefore order temperature" should be " Therefore the ordering temperature", etc.

In general, thanks to the neat experimental analysis and the clear demonstration of some rich aspects of the quantum physics of a higher-spin compound series, I am oriented to recommend the manuscript for publication in Nature Communication, after the authors address the above comments.

Reply to reviewers

Reviewer #1 (Remarks to the Author):

The manuscript "Bose-Einstein Condensations in Diluted $S = 3/2$ Quantum Magnets: Evolution from single- to double-dome structure" by Watanabe and coworkers reports on the properties of a BEC system with a spin of $3/2$. The probe used to explore the BEC phase diagram is the level of magnetic dilution in $\text{Ba}_2\text{CoGe}_2\text{O}_7$ where Co is partially exchanged with Zn leading to theoretically predictable changes. The authors find in addition, that magnetic dilution is leading to quenched disorder and percolation in $\text{Ba}_2\text{Co}_{1-x}\text{Zn}_x\text{Ge}_2\text{O}_7$ as a result of the formation of magnetic clusters thereby suppressing long-range magnetic order.

The authors went to great length to measure the temperature-magnetic field phase diagram of $\text{Ba}_2\text{Co}_{1-x}\text{Zn}_x\text{Ge}_2\text{O}_7$ single crystals ($x = 0, 0.125, 0.25, 0.375, 0.5$) using magnetostriction and electric permittivity in magnetic fields up to 50T. Their reported findings are supported and confirmed by theoretical model calculations. It is also interesting to note that the theoretical discussion can be extended to spin systems with $S > 2$ and strong easy-plane anisotropy. This will certainly help to gain a good understanding of the magnetic properties of BEC systems in general.

Overall the manuscript is well presented, the experiment and their findings explained clearly and the results are consistent in particular that the theoretical modelling seems to work well. I can therefore recommend the manuscript for publication in NCOMMS.

We greatly acknowledge Reviewer #1 for the positive evaluation and recommendation.

Reviewer #2 (Remarks to the Author):

General comments

This is an interesting paper although I am not convinced it meets the criterion of quality and originality expected for Nature Communications.

The authors explore the effects of magnetic field on a spin- $3/2$ antiferromagnet with weak planar anisotropy. They use doping as a way to control the exchange interactions and observe 2

connected domes of Bose-Einstein condensates as a function of field. Of course, an order state with gapless excitation is a trivial example of a Bose-Einstein condensate which is routinely measured. However it is possible to use a magnetic field to tune a system out of this phase by making the excitations gapped. In the case of $\text{spin} > 1$ and planar anisotropy which separate the spin components in energy, it is possible to enter further BEC phases as the higher spin states can condense into the ground state with increasing field.

The authors show this process in a qualitative way. They use a spin-3/2 magnet which orders magnetically and has a single BEC dome due to the strong exchange interactions between the magnetic ions. By doping they are able to control the exchange interactions allowing them to explore the weak interaction regime where 2 domes of Bose-Einstein condensates are observed as a function of field.

We thank Reviewer #2 who found our work interesting. We have carefully addressed all the concerns raised, and revised the manuscript.

I call their results qualitative because strictly speaking one needs to confirm the correct critical exponent to prove BEC whereas their exponents are substantially different probably due to disorder effects from doping which they indeed discuss. Furthermore they do not actually see the expected 2 domes, rather their BEC domes are joined. Greater doping might separate the domes better as a function of field but would introduce more disorder, hence this is not further explored in detail.

As Reviewer #2 points out correctly, unlike clean systems, we cannot rely on exponents to prove conventional BEC physics in the presence of disorder, and the existence of the disorder makes quantitative treatment difficult. We partially overcome the difficulty of quantitative discussion about the disordered systems by using effective spin $S = 1$ models. For example, as discussed in the main text, the experimental result that two AFM domes merged below 1 K in $x = 0.25$ sample is theoretically anticipated from the ground state phase diagram of diluted $S = 1$ spin systems obtained from a quantum Monte Carlo simulation. The same model predicts the complete separation of two domes for $x \gtrsim 0.35$. To further solidify the theoretical consideration,

we have performed additional measurements for $x = 0.375$ (Fig. R1). Low-temperature permittivity measurements in pulsed fields provide the re-emergence of the second dome in this doping. This result is described in Section II and new Fig. S2 of the revised Supplementary Information. This is fully consistent with the theoretical consideration. In this way, we quantitatively show how the phase diagram evolves as a function of x .

Altogether the paper presents an interesting idea but it would be better to observe it in a clean undoped system with weaker interactions, where the 2 BEC domes are naturally separated and can be explored quantitatively. It would be good if the authors gave a full justification of their reasoning behind the choice of material for this study.

We agree that one needs to determine exponents to prove BEC in quantum magnets experimentally, and from that perspective, it may be more desirable to use a clean system as a platform for studying BEC in the $S = 3/2$ system if our only purpose is to show the BEC physics in a higher spin system. However, even if there were already studies of $S = 3/2$ BEC in a clean system, our study still had great importance and high originality, as our main focus is not just on the $S = 3/2$ BEC physics but mainly on revealing how the featureless single AFM phase evolves into two AFM domes. By doing so, we can fully comprehend the precise nature of higher spin BEC and give a hint to the challenging task of the model material search. Therefore, the engineering of J or D is the central point in our study. To make this point clearer, we modify the following sentences.

OLD

Accordingly, it requires an exquisite energy balance between J and D to realize this type of BEC phenomenon, which has challenged the experimental realization.

NEW

Accordingly, an exquisite energy balance between J and D is required to realize this type of BEC phenomenon, which has challenged the experimental realization. In order to establish a design principle for the higher spin BEC physics, we need to investigate how the featureless single AFM phase in the strongly interacting regime evolves into two AFM domes in the weakly interacting regime as a function of J/D .

Also, we would like to raise two primary reasons why we chose $\text{Ba}_2\text{CoGe}_2\text{O}_7$. (1) Magnetoelec-

tric coupling in $\text{Ba}_2\text{CoGe}_2\text{O}_7$ is extremely useful for detecting the phase transition in the magnetic sector even in the presence of disorder. (2) $\text{Ba}_2\text{CoGe}_2\text{O}_7$ is in the strongly-interacting regime but has a relatively strong D . Therefore, we expect that the magnetic order in the intermediate field range can easily be suppressed by reducing J , which enables the experimental demonstration of how the $S = 3/2$ system potentially shows BEC physics. (1) had already been mentioned in the text. But (2) had yet to be emphasized. Therefore, we modify the following sentences to stress (2).

OLD

AFM long-range order appears below $T_N = 6.7$ K at a zero field, and the magnetic phase diagram of BCGO consists of one broad AFM phase, indicating that this material is in the strongly-interacting regime [Fig. 1 (b)].

NEW

AFM long-range order appears below $T_N = 6.7$ K at zero field, and the magnetic phase diagram of BCGO consists of one broad AFM phase, indicating that this material is in the strongly-interacting regime [Fig. 1 (b)]. Therefore, BCGO is ideal for observing the transition from the strongly interacting regime to the weakly interacting regime.

Finally, we note that the randomness effect gives rise to new physics such as a dilution-induced quantum phase transition and the appearance of non-trivial quantum phases. These features do not exist in clean systems and are worth exploring in future studies.

Other comments

Page 3

'The anomaly becomes blunt' -> 'The anomaly becomes broadened' 'suggesting the weak exchange striction in BCZGO partly due to the long exchange path' -> 'suggesting that the exchange striction in BCZGO is weak partly due to the long exchange path' 'but does not in accordance with the finite temperature phase boundary' -> 'but does not agree quantitatively due to the finite temperature phase boundary'

Fig 2

It would be clearer if parts a) to c) which describe the basic structural properties were in a different figure from parts d) to i)

We appreciate the referee's suggestions which were useful for improving the manuscript. We

have revised the manuscript as suggested.

Reviewer #3 (Remarks to the Author):

The authors study Bose-Einstein condensation (BEC) in the compound series $\text{Ba}_2\text{Co}_{1-x}\text{Zn}_x\text{Ge}_2\text{O}_7$. The essential physics is captured by a relatively simple model of $S = 3/2$ spins on the square lattice with easy-plane single ion anisotropy D , interacting via the Heisenberg exchange interaction. When D is small compared to the exchange coupling J , the antiferromagnetic region in the longitudinal magnetic field vs. temperature phase diagram is featureless. However, when J is lowered a double-peak structure of the same phase appears. In the limiting case of very small J the double peak structure actually evolves into two separate domes that can be seen as two different Bose-Einstein condensates. This structure is richer than the single BEC dome that appears in most of the previous theoretical and experimental works, though it could be somewhat reminiscent of that of spin-1 couple dimer compounds (see $\text{Ba}_3\text{Mn}_2\text{O}_8$).

The Hamiltonian that is supposed to describe $\text{Ba}_2\text{CoGe}_2\text{O}_7$ more accurately is actually more complicated and, in particular, includes a certain form of magnetoelectric coupling; the latter is what allows to use the measurements of the electric permittivity to detect the magnetic phase transitions (more conventional magnetic measurements are also performed). The big theoretical assumption is that the same kind of Hamiltonian describes the chemically substituted compounds, with the only difference that the ratio J/D becomes smaller; in other words, the dilution of the magnetic sites is taken into account phenomenologically by decreasing J in the same Hamiltonian. This proves to be a reasonable assumption by looking at the fair agreement between the experimental data and the mean-field calculation based on such theoretical model. As expected, there are features that cannot be explained within this framework, such as the critical exponents (related to the shape of the phase transition line) near the quantum critical point; in principle it should be possible to do a more systematic analysis of the effect of disorder to deal with this aspect, but it is fair to say that it goes beyond the scope of this paper and the authors give sensible arguments based on the existing literature.

We thank Reviewer #3 for the positive comments. We agree with Reviewer #3. In particular, we think that $\text{Ba}_3\text{Mn}_2\text{O}_8$, also cited in the manuscript, is the couple dimer counterpart of our study. We have carefully addressed all the raised concerns and revised the manuscript.

I would like the authors to address the following points.

i) By looking at the various figures, I am under the impression that the authors have enough data to draw the H - T phase diagram also for $x = 0.375$. If this is true, it would be nice to show that phase diagram as well. Even if the nature of the non-magnetic phase at low T remains somewhat elusive as discussed in the text, that could be indicated in the caption and/or the figure.

We thank Reviewer #3 for the constructive comments. In the first submission, we refrained from drawing the magnetic phase diagram for $x = 0.375$ because it was too speculative at that stage, even though we could infer the shape of the dome from $x = 0.25$ data and theoretical considerations used in the manuscript. To resolve this issue, we performed an additional measurement of electric permittivity down to 0.7 K in a pulsed magnetic field for $x = 0.375$ and presented the updated phase diagram in the Supplementary Information.

Figure R1 (a) shows the magnetic-field dependence of electric permittivity ϵ_a . We can see the enhancement of ϵ_a around 22 T, which highly suggests the existence of the second dome. Therefore now we can discard the scenario that $x = 0.375$ just has a single AFM dome in the low field region. Furthermore, using the same definition used in $x = 0.25$, we add two points in the phase diagram for $x = 0.375$, as indicated by squares in Fig. R1 (b). Similarly to the $x = 0.25$ case, we expect that the splitting of the peaks which indicates critical fields would become clearer at lower temperatures. As discussed in the Supplementary Information, in order to draw a solid conclusion about the qualitative values of the second dome, we need further experiments, such as the electric permittivity measurements in a static magnetic field, or, as Reviewer #3 carefully suggested, a further systematic study of the x dependence, which is beyond the scope of the current study.

ii) The authors should make a little clearer that, while they use Eq.(1) as a pedagogical model to explain the essential physics, as soon as they start to deal more concretely with $\text{Ba}_2\text{Co}_{1-x}\text{Zn}_x\text{Ge}_2\text{O}_7$ they actually use Eq.(5). I think that would actually make it easier for the reader to immediately appreciate the rationale and the value of the permittivity measurements.

We thank Reviewer #3 for the constructive comments. We add a sentence below just after

introducing Eq. (1) so that the readers can recognize a minor difference between the ideal model and the actual material.

” The existence of $\hat{\mathcal{H}}_{\text{DM}}$ and $\hat{\mathcal{H}}_{\text{me}}$ in $\text{Ba}_2\text{CoGe}_2\text{O}_7$ (BCGO), the material studied in this paper, allows sensitive detection of a magnetic order through electric measurements [22]. Still, as far as these two terms are small compared with the first three terms, which is the case in BCGO [23], the BEC physics would remain qualitatively unchanged from the case of the no-magnetoelectric coupling. Therefore, hereafter, we will continue the discussion without considering $\hat{\mathcal{H}}_{\text{DM}}$ and $\hat{\mathcal{H}}_{\text{me}}$. See Methods for the explicit form of the magnetoelectric coupling in BCGO. ”

iii) I would recommend to thoroughly check the English; e.g., ”Therefore order temperature” should be ” Therefore the ordering temperature”, etc.

We thank Reviewer #3 for the suggestion. The English is checked by a native English speaker and corrected properly, as highlighted in brown in the main text.

In general, thanks to the neat experimental analysis and the clear demonstration of some rich aspects of the quantum physics of a higher-spin compound series, I am oriented to recommend the manuscript for publication in Nature Communication, after the authors address the above comments.

We thank again Reviewer #3 for evaluating our work.

[R1] J. W. Kim, S. Khim, S. H. Chun, Y. Jo, L. Balicas, H. T. Yi, S. W. Cheong, N. Harrison, C. D. Batista, J. Hoon Han, and K. Hoon Kim, Nature Communications **5**, 4419 (2014).

Fig. R1. **Extended magnetic phase diagram with possible phase boundaries for $x = 0.375$.** (a) Magnetic-field dependence of electric permittivity ϵ_a and its field derivative $d\epsilon_a/dH$ for $x = 0.375$. The measurements at 0.7 K were performed twice, and we show both $d\epsilon_a/dH-H$ curves, which are reproducible. The local minimum and maximum in the $d\epsilon_a/dH-H$ curve used to obtain the critical fields H_{c2} and H_{c3} are shown by purple triangles (see Supplemental Information for the definition of critical fields). (b) Magnetic phase diagram for $x = 0, 0.25$ and 0.375 . The black circles correspond to the phase boundary of pristine $\text{Ba}_2\text{CoGe}_2\text{O}_7$, which are collected from Ref. [R1].

Summary of changes

The English is checked by a native English speaker and corrected properly, as highlighted in brown in the main text. We have added chapter labels such as Results and Discussion, also highlighted in brown. All the other changes are highlighted in red and listed below.

- Split originally single Figure 2 into two figures (Figure 2 and Figure 3).
- Add the section about the definition of the critical fields H_{ci} ($i = 1, 2, 3$) to the Supplementary Information.
- Revise Section III of the revised Supplementary Information to clarify the contents.
- Add the experimental data for $x = 0.375$ and extended phase diagram to the Supplementary Information.
- Equation (1). Add the Dyallosinskii-Moriya term and magnetoelectric coupling term to the Hamiltonian.
- Page 1, column 2, line 11.

NEW

The existence of $\hat{\mathcal{H}}_{\text{DM}}$ and $\hat{\mathcal{H}}_{\text{me}}$ in $\text{Ba}_2\text{CoGe}_2\text{O}_7$ (BCGO), the material studied in this paper, allows sensitive detection of a magnetic order through electric measurements [22]. Still, as far as these two terms are small compared with the first three terms, which is the case in BCGO [23], the BEC physics would remain qualitatively unchanged from the case of the no-magnetoelectric coupling. Therefore, hereafter, we will continue the discussion without considering $\hat{\mathcal{H}}_{\text{DM}}$ and $\hat{\mathcal{H}}_{\text{me}}$. See Methods for the explicit form of the magnetoelectric coupling in BCGO.

- Page 2, column 1, line 32.

OLD

Accordingly, it requires an exquisite energy balance between J and D to realize this type of BEC phenomenon, which has challenged the experimental realization.

NEW

Accordingly, an exquisite energy balance between J and D is required to realize this type of BEC phenomenon, which has challenged the experimental realization. In order to establish

a design principle for the higher spin BEC physics, we need to investigate how the featureless single AFM phase in the strongly interacting regime evolves into two AFM domes in the weakly interacting regime as a function of J/D .

- Page 2, column 2, line 26.

OLD

AFM long-range order appears below $T_N = 6.7$ K at a zero field, and the magnetic phase diagram of BCGO consists of one broad AFM phase, indicating that this material is in the strongly-interacting regime [Fig. 1 (b)].

NEW

AFM long-range order appears below $T_N = 6.7$ K at a zero field, and the magnetic phase diagram of BCGO consists of one broad AFM phase, indicating that this material is in the strongly-interacting regime [Fig. 1 (b)]. Therefore, BCGO is ideal for observing the transition from the strongly interacting regime to the weakly interacting regime.

- Add the descriptions about spin-dependent electric polarization P_i in BCGO to the "Method" section.

REVIEWERS' COMMENTS

Reviewer #2 (Remarks to the Author):

The authors have answered most of my original questions and they are able to explain well the phenomena that they observe in BCZGO, the paper has definitely improved.

One final question I have concerns the magnetic structure shown in figure 2b which is a non-collinear structure. How does this collinearity which appear rather strong in the figure arise from the Hamiltonian Eq (1) (could this be second neighbour or the DM term so strong)? And what are the consequences of the dilution process on this non-collinearity.

Reviewer #3 (Remarks to the Author):

Having examined all the newly available material, I feel that the authors have carefully addressed my previous comments and, more in general, the manuscript has seen significant improvements.

Incidentally, I happened to notice the misspelling of one of the authors' names, which should be "Kenichiro Hashimoto", if I am not mistaken.

Given the present version of the manuscript, I think that the robust experimental work, accompanied by the new, more convincing presentation, deserves publication in Nature Communications.

Reply to reviewers

Reviewer #2 (Remarks to the Author):

The authors have answered most of my original questions and they are able to explain well the phenomena that they observe in BCZGO, the paper has definitely improved.

We greatly acknowledge Reviewer #2 for the positive evaluation of our revised manuscript.

One final question I have concerns the magnetic structure shown in figure 2b which is a non-collinear structure. How does this collinearity which appear rather strong in the figure arise from the Hamiltonian Eq (1) (could this be second neighbour or the DM term so strong)? And what are the consequences of the dilution process on this non-collinearity.

As Reviewer #2 pointed out, the magnetic moments of Co ions are canted in the *ab*-plane from the collinear antiferromagnetic order due to the in-plane Dzyaloshinskii-Moriya interaction. Previous polarized neutron diffraction study suggests that the canting angle is less than 0.2 [1]. In Fig. 2b, we show the schematic of the magnetic structure and exaggerate the canting of the moment from the actual angle for visibility. In BCGO with magnetic ordering vector $\mathbf{k} = (1\ 0\ 0)$, zero-field spin canting causes a weak spontaneous magnetization in field dependence of magnetization [2]. As shown in Supplementary Fig. 1, we confirmed the residual magnetization at zero-field for BCZGO with $x = 0.25$ suggesting the small canting of Co ions moments. Therefore, we conclude that the dilution process remains the zero-field magnetic structure intact.

Reviewer #3 (Remarks to the Author):

Having examined all the newly available material, I feel that the authors have carefully addressed my previous comments and, more in general, the manuscript has seen significant improvements. Incidentally, I happened to notice the misspelling of one of the authors' names, which should be "Kenichiro Hashimoto", if I am not mistaken. Given the present version of the

manuscript, I think that the robust experimental work, accompanied by the new, more convincing presentation, deserves publication in Nature Communications.

We greatly acknowledge Reviewer #3 for the positive evaluation and recommendation for the publication. We also appreciate your appropriate correction about the author's name.

-
- [1] V. Hutanu, A. P. Sazonov, M. Meven, G. Roth, A. Gukasov, H. Murakawa, Y. Tokura, D. Szaller, S. Bordács, I. Kézsmárki, V. K. Guduru, L. C. J. M. Peters, U. Zeitler, J. Romhányi, and B. Náfrádi, Evolution of two-dimensional antiferromagnetism with temperature and magnetic field in multiferroic $\text{Ba}_2\text{CoGe}_2\text{O}_7$, *Physical Review B* **89**, 064403 (2014).
- [2] H. T. Yi, Y. J. Choi, S. Lee, and S. W. Cheong, Multiferroicity in the square-lattice antiferromagnet of $\text{Ba}_2\text{CoGe}_2\text{O}_7$, *Applied Physics Letters* **92**, 212904 (2008).